# The last resort antibiotic daptomycin exhibits two independent antibacterial mechanisms of action

Jessica A. Buttress[1], Ann-Britt Schäfer[2,3], Alan Koh [1,4,5], Jessica Wheatley [1], Katarzyna Mickiewicz [1], Michaela Wenzel [2,3] & Henrik Strahl [1] ✉

Daptomycin is a lipopeptide antibiotic, commonly used as last resort treatment against multidrug resistant Gram-positive pathogens. Despite its clinical success, the mechanism through which daptomycin exerts its antibacterial properties has remained controversial. Much of the debate is focused around daptomycin's ability to depolarise the cytoplasmic membrane, potential formation of large membrane pores, and its more recently discovered capability to inhibit cell wall synthesis and disturb membrane lipid domain organisation through interactions with cell wall precursor lipids. Here we show that, rather than representing different facets of a single underlying activity, daptomycin exhibits two independent antibacterial mechanisms of action: (i) targeting of cell wall precursor lipids that leads to cell wall synthesis inhibition and (ii) membrane depolarisation that does not rely on interactions with cell wall precursor lipids. This dual mechanism of action provides an explanation for the frequently disagreeing findings obtained through in vivo and in vitro studies and explains why resistance development towards daptomycin is slow and multifactorial. In a broader context, this demonstrates that dual mechanism antibiotics can be clinically successful and exhibit favourable characteristics in terms of real-world, clinically relevant resistance development.

Daptomycin is a lipopeptide antibiotic that displays excellent activity against Gram-positive pathogens and is often implemented as a last resort agent to combat multidrug-resistant bacterial infections[1]. Despite its clinical importance, the precise mechanism of action of daptomycin is still heavily debated. In particular, much controversy is centred on whether daptomycin forms large pores in the cytoplasmic membrane of target cells and how it inhibits cell wall synthesis; with many discrepancies transpiring from differences between in vivo and in vitro studies[2–4].

Daptomycin, originally isolated as a secondary metabolite produced by the Gram-positive soil bacterium *Streptomyces roseosporus*[5,6], is a lipodepsipeptide consisting of a 10-membered cyclic lactone core and three exocyclic amino acids linked to a decanoyl fatty acid tail[7,8]. Unlike most antibacterial lipopeptides, daptomycin is negatively charged, and its antimicrobial activity depends on bound $Ca^{2+}$ ions, which reduce the negative charge of the peptide and stimulate its oligomerisation[9–12]. The resulting daptomycin-calcium complex has an increased affinity for the anionic bacterial phospholipid phosphatidylglycerol (PG) compared to other bacterial phospholipid species[13].

[1]Centre for Bacterial Cell Biology, Biosciences Institute, Faculty of Medical Sciences, Newcastle University, Newcastle upon Tyne, UK. [2]Division of Chemical Biology, Department of Life Sciences, Chalmers University of Technology, Gothenburg, Sweden. [3]Center for Antibiotic Resistance Research in Gothenburg (CARe), Gothenburg, Sweden. [4]MRC Laboratory of Medical Sciences, London, UK. [5]Institute of Clinical Sciences, Imperial College, London, UK. ✉e-mail: h.strahl@ncl.ac.uk

PG is particularly abundant in the cell membranes of Gram-positive bacteria[14], thereby promoting daptomycin's selectivity for bacterial over mammalian membranes[15]. Gram-negative bacteria are generally resistant towards daptomycin due to its inability to traverse the outer membrane barrier[16], although their different phospholipid composition also appears to play a role[17].

Studies focusing on daptomycin's mechanism of action, both in terms of direct drug-target interactions leading to inhibition and cellular consequences of target inhibition (frequently termed mode of action), have provided controversial results for several decades, with the earliest research arguing that daptomycin inhibits peptidoglycan synthesis[7,18]. The drug approval by FDA, in contrast, cites membrane depolarisation as the primary mechanism of action[19]. Previous work by ourselves and collaborators demonstrated, using fluorescent lipid probes and fluorescent protein fusions, that daptomycin preferentially inserts into fluid membrane microdomains in *Bacillus subtilis* and triggers their clustering[20]. This change in membrane organisation impairs the attachment of several peripheral membrane proteins, most prominently the lipid II synthase MurG, the membrane association of which is essential for peptidoglycan synthesis[21]. More recently, daptomycin was demonstrated to inhibit the cell wall synthetic machinery even more directly by forming a tripartite complex with PG and undecaprenyl-coupled cell wall precursors including lipid II[22,23]. The associated steric hindrance likely explains the observed inhibition of cell wall synthesis[7,18], as well as dissociation of MurG from the membrane surface[20]. The question remains, however, why daptomycin remains active against both *Enterococcus faecium* protoplasts and cell-wall-less *B. subtilis* L-forms, which both lack a cell wall and therefore are unlikely to be affected by the cell wall-targeting activity of this antibiotic[24,25]. In contrast to a mechanism based on inhibition of cell wall synthesis, several in vitro studies using model membrane systems have shown that daptomycin forms cation-selective, relatively small membrane pores of less than 10 Å diameter[26,27]. When tested in vivo, comparatively high daptomycin concentrations or prolonged incubation times were required to observe membrane depolarisation in *B. subtilis* and *Staphylococcus aureus*[20,28], whilst no ion leakage was observed in *B. subtilis* at low yet growth-inhibitory concentrations[3,20]. This is in clear contrast to other pore-forming compounds that act rapidly[29,30]. Lastly, a delayed response to membrane permeability indicator dyes has been observed in daptomycin-treated *B. subtilis*, which was interpreted as the formation of large membrane pores[31]. However, *B. subtilis* undergoes extensive autolysis in response to conditions that compromise membrane integrity or cell wall synthesis including incubation with daptomycin[32–34], thus complicating interpretation of the delayed membrane effects of daptomycin.

In this study, we observed that daptomycin indeed triggers extensive autolysis in the Gram-positive model organism *B. subtilis* under conditions that were previously linked to membrane pore formation. In *B. subtilis* strains deficient for key peptidoglycan-degrading enzymes required for induced autolysis, or in *S. aureus* that natively does not undergo autolysis in response to daptomycin, no significant membrane pore formation was observed. Thus, rather than representing daptomycin's direct membrane activity, the delayed effects of daptomycin on *B. subtilis* membrane integrity are secondary and caused by the cell's autolytic processes. Using L-form cells lacking cell wall precursor lipids, we demonstrate here that daptomycin exhibits two distinct and independent mechanisms of action: (i) interaction with cell wall precursor lipids resulting in clustering of lipid II and associated fluid membrane microdomains and (ii) depolarisation of the cytoplasmic membrane that is independent of interactions with cell wall precursor lipids. Finally, we show that rather than being due to direct daptomycin-membrane interaction, the large membrane domain clusters observed in *B. subtilis* in response to daptomycin are formed as part of the Lia-stress response and specifically require the IM30 protein LiaH[35–37].

## Results

### Daptomycin induces autolysis without pore formation in *B. subtilis*

The ability of daptomycin to induce membrane permeabilisation through pore formation has remained controversial[3]. In vivo research on this key question regarding daptomycin's mode of action has largely overlooked the cell wall autolytic process[31,32], which complicates the interpretation of in vivo membrane permeability assays. To investigate whether daptomycin indeed forms large pores in the cytoplasmic membrane of *B. subtilis* and the kinetics of the process, we performed time-resolved SYTOX Green membrane permeability assays using a fluorescence plate reader. SYTOX Green is a membrane-impermeant DNA intercalating dye that can stain the bacterial nucleoid when cell entry is facilitated by pores formed within the cytoplasmic membrane[38]. Given the size of the SYTOX Green molecule (~600 Da), a large pore size is needed for its passage. Indeed, the known pore-forming lantibiotic nisin induced a rapid increase in SYTOX Green fluorescence (Fig. 1a). Daptomycin treatment (at 6-times MIC, Supplementary Table 1) also increased SYTOX Green staining; however, consistent with previous studies using a similar permeability indicator dye propidium iodide[31], this was found to be a slow process starting ~30 min following daptomycin addition. Due to this delay, and the ability of membrane-targeting compounds to induce autolysis in *B. subtilis*[33], we tested whether this apparent membrane permeation was due to autolysis or direct membrane pore formation. For this aim, the assay was repeated with a *B. subtilis* strain deficient for the major autolysins (Δ*lytABCDEF*). The minimum inhibitory concentration (MIC) of daptomycin was not affected by the absence of autolysins (1.6 µg/ml in wild-type and Δ*lytABCDEF* cells, Supplementary Table 1), yet a clear suppression in daptomycin-induced SYTOX Green fluorescence was observed in this strain, indicating involvement of autolytic processes (Fig. 1b). Similar suppression was not observed for nisin.

To observe how this population-based data translates to the behaviour of individual cells, and to assess the potential cell-to-cell heterogeneity involved, phase contrast and SYTOX Green fluorescence microscopy was performed on *B. subtilis* wild-type. Cells were treated with two concentrations of daptomycin (the minimum concentration to inhibit *B. subtilis* growth at an OD$_{600}$ of 0.3 [Supplementary Fig. 1] and twofold this), and at two time points (5 min and 25 min), to observe the early and late stages of its action. It became immediately apparent, especially at the later time point (25 min), that many of the cells had indeed lysed as indicated by the presence of phase light cells (white arrows; Fig. 1c). In fact, when the proportion of phase light cells was quantified, irrespective of the used daptomycin concentration, ~30–40% of all cells had lysed by 25 min of treatment (Fig. 1d). Importantly, the majority of the remaining phase dark cells were not stained by SYTOX Green, in contrast to cells incubated with nisin. We therefore hypothesised that, rather than large pore formation allowing SYTOX Green to enter the cell, the delayed SYTOX Green fluorescence observed in the plate reader data was due to staining of extracellular DNA released during autolysis. Indeed, lysis-derived extracellular DNA stained by SYTOX Green was frequently observed in cells incubated with daptomycin, whilst nisin-treated cells consistently exhibited intracellular staining indicative of pore formation (Fig. 1e). Finally, we validated that daptomycin was binding to cells by exploiting the fluorescence emission from its naturally fluorescent amino acid kynurenine[10,39] (Fig. 1e). In conclusion, whilst daptomycin can bind *B. subtilis* cells and trigger positive SYTOX Green signals, these appear to be linked to cell autolysis rather than being indicative of pore formation.

### Daptomycin does not trigger pore formation even in the absence of autolysis

As the majority of cells were undergoing cell lysis at the later time points, we next investigated the action of daptomycin in cells

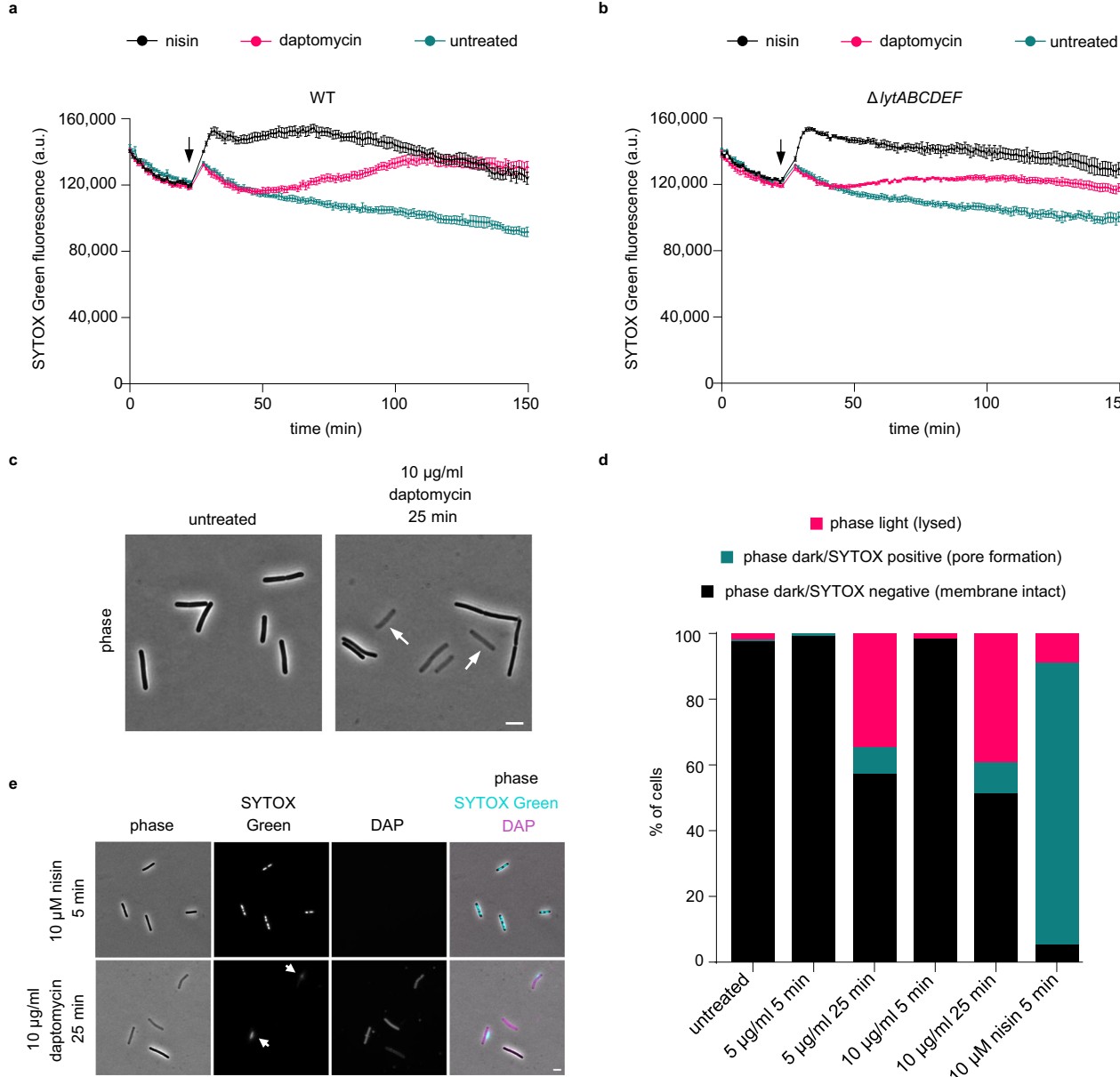

**Fig. 1 | Daptomycin induces lysis without significant membrane pore formation in *B. subtilis*.** The effect of 10 μg/ml daptomycin on the fluorescence intensity of 1 μM SYTOX Green in **a** *B. subtilis* wild-type cells and **b** *B. subtilis* cells deficient for cell wall autolytic proteins, LytA-F, at an OD$_{600}$ of 0.5. The pore-forming lantibiotic nisin (10 μM) was used for positive control. The time point of antibiotic addition is indicated by an arrow. Data are represented as mean ± SD of three technical replicates. See Supplementary Fig. 2 for similar data obtained with *S. aureus*. **c** Phase contrast microscopy images of *B. subtilis* wild-type cells untreated or treated with 10 μg/ml daptomycin for 25 min. Fully lysed (phase light) cells are highlighted with white arrows. **d** *B. subtilis* wild-type cells were stained with 200 nM SYTOX Green and treated with either 5 μg/ml or 10 μg/ml daptomycin for 5 and 25 min.

Percentage of total cells phase light (lysed), phase dark/SYTOX positive (pore formation) or phase dark/SYTOX negative (membrane intact) are shown. SYTOX positive was defined as a fluorescence intensity of 5000 a.u. or greater. 10 μM of the lantibiotic nisin was used as a positive control for pore formation. *n* = 116–147. **e** Phase contrast and fluorescence microscopy images of *B. subtilis* wild-type cells with SYTOX Green and daptomycin shown in cyan and magenta, respectively, and treated with either 10 μM nisin or 10 μg/ml daptomycin. White arrows indicate staining of extracellular DNA released during autolysis. Daptomycin fluorescence was detected based on kynurenine autofluorescence (DAP). Scale bar, 3 μm. Strains used: *B. subtilis* 168 (WT) and *B. subtilis* KS19 (Δ*lytABCDEF*). For details about the sample size, see Supplementary Table 2.

lacking key cell wall-lytic enzymes and, thus, exhibiting suppressed autolytic behaviour. For this aim, we performed phase contrast and SYTOX Green fluorescence microscopy of daptomycin-treated *B. subtilis* Δ*lytABCDEF*. Deletion of the major autolysins in *B. subtilis* results in a cell chaining phenotype linked to a defect in cell separation after division[40]. To distinguish individual cells within the chain, the cells were co-stained with the membrane dye FM 5-95 (Fig. 2a). As expected, daptomycin-induced autolysis observed for the wild-type cells was abolished in *B. subtilis* Δ*lytABCDEF* (Fig. 2).

Crucially, most cells remained free from SYTOX Green staining at both daptomycin concentrations and time points, compared to ~100% of nisin-treated cells exhibiting a positive, intracellular SYTOX Green signal (Fig. 2b). Again, we confirmed binding of the antibiotic to cells through detection of daptomycin's intrinsic fluorescence (Fig. 2c). Overall, this demonstrates that, even in the absence of autolysis that might obscure and overshadow other more direct activities, no large pore formation occurs in *B. subtilis* as a result of daptomycin treatment.

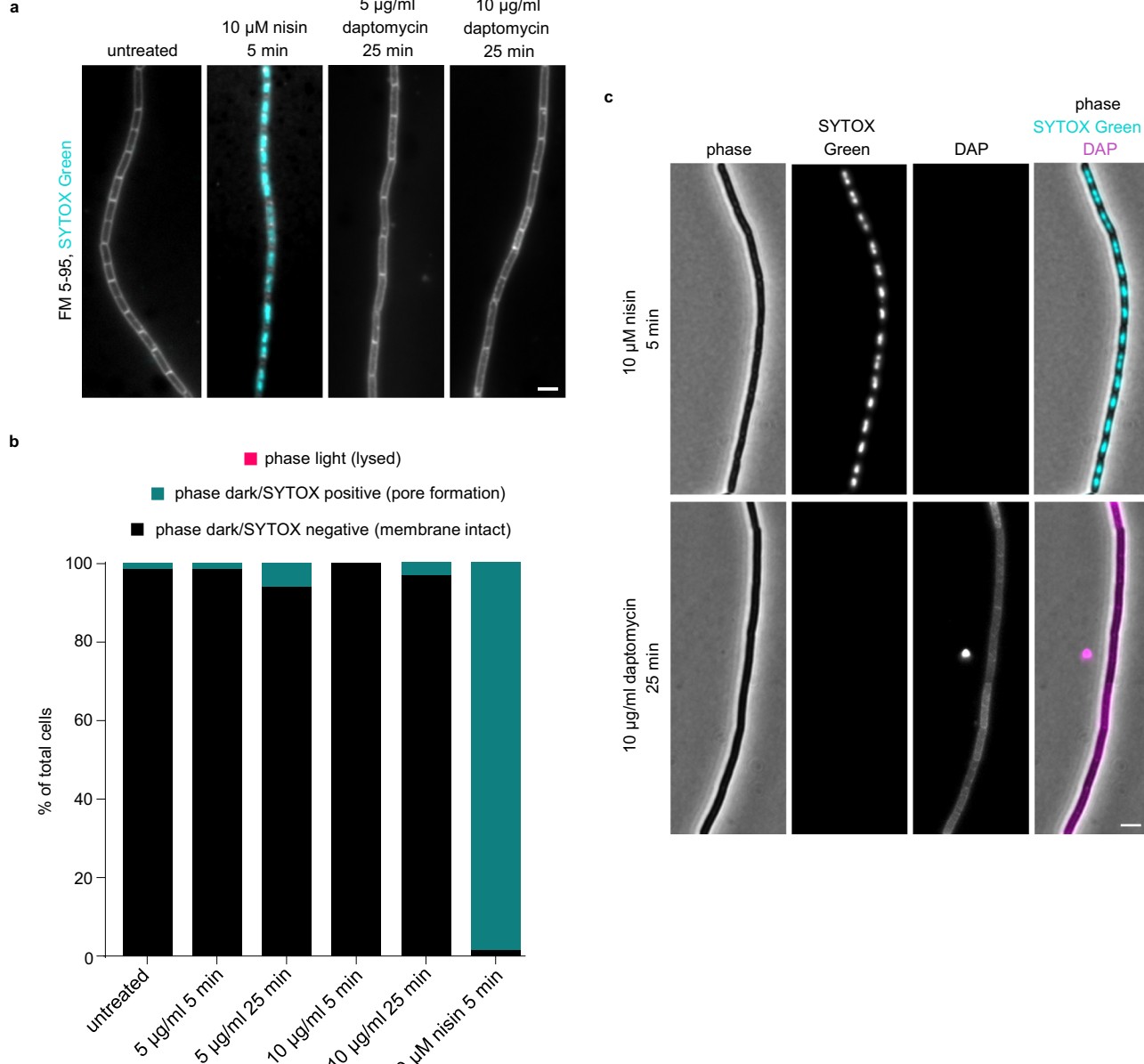

**Fig. 2 | Daptomycin does not trigger membrane pore formation even in the absence of autolysis. a** Fluorescence microscopy images of *B. subtilis* Δ*lytABCDEF* cells co-stained with 2 µg/ml FM 5-95 (shown in grey) and 200 nM SYTOX Green (shown in cyan), in the absence or presence of either 5 µg/ml or 10 µg/ml daptomycin for 5 and 25 min (only the later timepoint shown). **b** Quantification of the percentage of phase light (lysed), phase dark/SYTOX positive (pore formation) or phase dark/SYTOX negative cells (membrane intact) is shown. 10 µM of the lantibiotic nisin was used as a positive control for pore formation. *n* = 101–144. **c** Phase contrast and fluorescence microscopy of *B. subtilis* Δ*lytABCDEF* cells, with SYTOX Green and daptomycin shown in cyan and magenta, respectively, and treated with either 10 µM nisin or 10 µg/ml daptomycin. Daptomycin fluorescence was detected based on kynurenine autofluorescence. Scale bar, 3 µm. Strain used: *B. subtilis* KS19 (Δ*lytABCDEF*). For details about the sample size, see Supplementary Table 2.

## Absence of pore formation upon daptomycin treatment is conserved in *Staphylococcus aureus*

As daptomycin is commonly used against multidrug-resistant Gram-positive infections[1], we decided to test whether this mode of action was also observed in the Gram-positive pathogen *S. aureus*. In contrast to *B. subtilis*, treatment with daptomycin at a range of concentrations from 1 to 20 µg/ml did not trigger lysis in *S. aureus* wild-type cells, even though growth was inhibited at concentrations above 10 µg/ml (Supplementary Fig. 1). To follow daptomycin's effect on membrane potential and integrity in *S. aureus*, we performed a combined DiSC$_3$(5) and SYTOX Green fluorescence microscopy time course experiment of cells treated with 5 µg/ml and 10 µg/ml daptomycin, (10-times and 20-times MIC, respectively; Supplementary Table 1). DiSC$_3$(5) is a voltage-sensitive fluorescent dye which, due to its cationic and hydrophobic nature, accumulates in polarised cells and is released upon depolarisation[41,42]. Similar to previous observations in *B. subtilis*[20], membrane depolarisation was gradual, heterogeneous, and concentration-dependent (Fig. 3a, b). Crucially, no membrane pore formation was observed even for cells that were substantially depolarised, with only a minor fraction of the cell population exhibiting positive SYTOX Green signals after an extended incubation time of 60 min (Fig. 3a, b). This is in clear contrast to the positive control nisin which acts through pore formation and, thus, exhibits depolarisation that coincides with positive SYTOX Green signals (Fig. 3a). Thus, the ability of daptomycin to trigger membrane depolarisation in a manner that does not rely

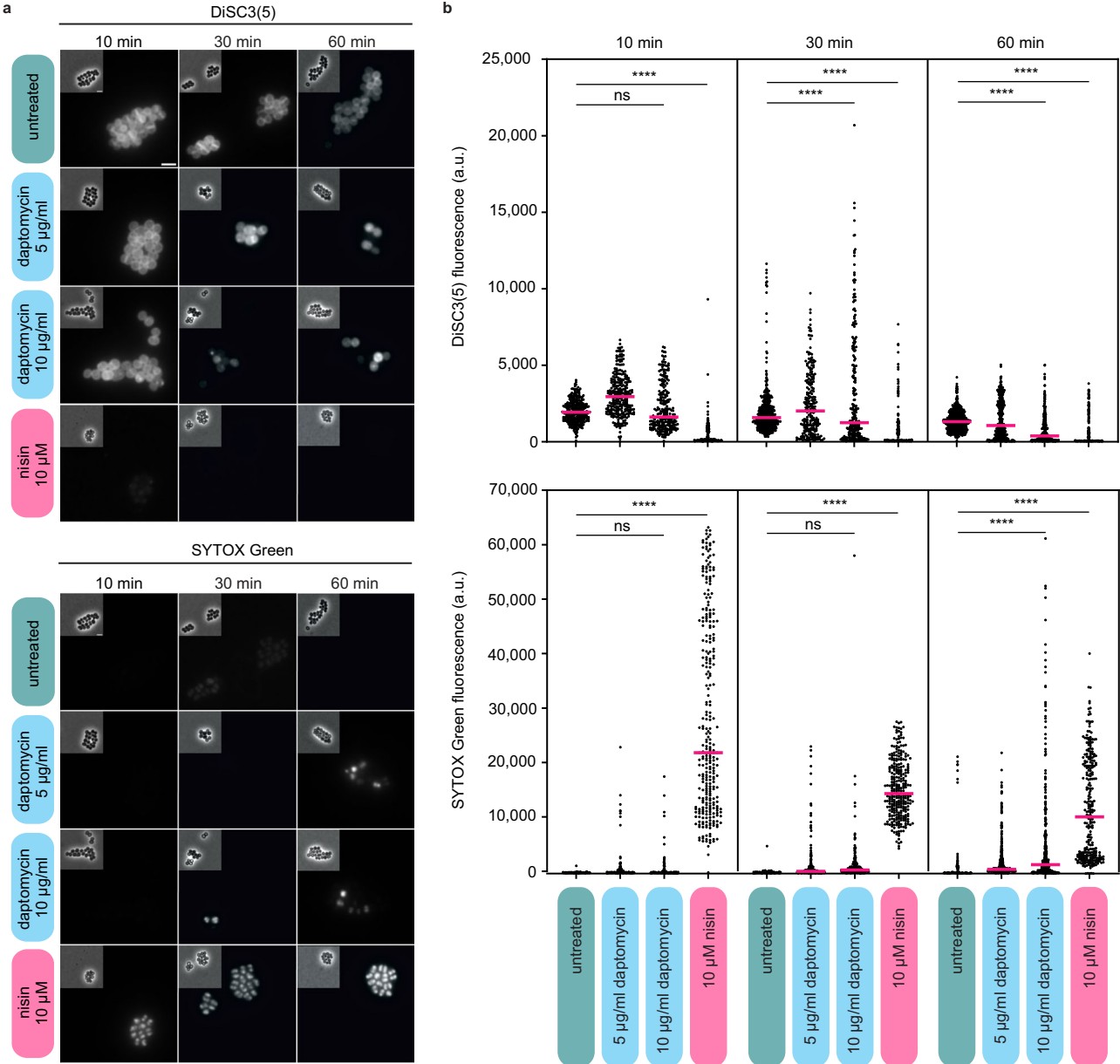

**Fig. 3 | Daptomycin induces a slow and incomplete membrane depolarisation in *S. aureus*, without membrane pore formation. a** Phase contrast and fluorescence microscopy images of *S. aureus* cells stained with 200 nM SYTOX Green and 150 nM DiSC$_3$(5) and treated with 5 μg/ml or 10 μg/ml daptomycin for 10, 30 and 60 min. The pore-forming lantibiotic nisin was used as a positive control for membrane pore formation and depolarisation. Scale bar, 2 μm. **b** Quantification of fluorescence for individual cells from the dataset shown in (**a**) (*n* = 231–706). Median fluorescence intensity is indicated with a magenta line, together with *P* values of a one-way, unpaired ANOVA. \*\*\*\* represents *p* < 0.0001, ns, not significant. Strain used: *S. aureus* SH1000 (WT). For details about the sample size and statistical significance, see Supplementary Table 2.

on large pore formation is conserved between *B. subtilis* and *S. aureus*.

**Daptomycin does not cluster fluid membrane microdomains in the absence of cell wall precursor lipids, but still triggers membrane depolarisation**

It was previously shown by ourselves and collaborators that daptomycin interferes with fluid membrane microdomains in *B. subtilis*; hypothesised to be due to induced clustering of cell wall precursor lipids such as lipid II[20]. More recently, a direct interaction between daptomycin, undecaprenyl-coupled peptidoglycan precursors and the anionic phospholipid PG was demonstrated by Grein et al.[22]; however, a later study using isothermal titration calorimetry was unable to demonstrate lipid II binding by daptomycin[13]. We therefore

aimed to investigate whether the fluid membrane microdomain clusters induced by daptomycin in vivo are indeed mediated by its interactions with cell wall precursor lipids. Since cell wall synthesis is essential for normal *B. subtilis* cells, we focused on L-forms, which are cell wall-deficient bacterial variants[43,44]. *B. subtilis* can be converted to an L-form state either by deleting essential genes in the cell wall biosynthesis pathway or by stimulating phospholipid synthesis in the presence of cell wall antibiotics[45]. To test the role of cell wall precursor lipids in the formation of fluid membrane microdomains, we deleted *uppS*, which encodes undecaprenyl pyrophosphate synthase, thereby eliminating the synthesis of any undecaprenyl-coupled cell wall precursors including lipid II. Perhaps surprisingly, daptomycin was equally growth-inhibitory in both L-forms that maintain synthesis of cell wall precursor lipids but do not synthesise peptidoglycan,

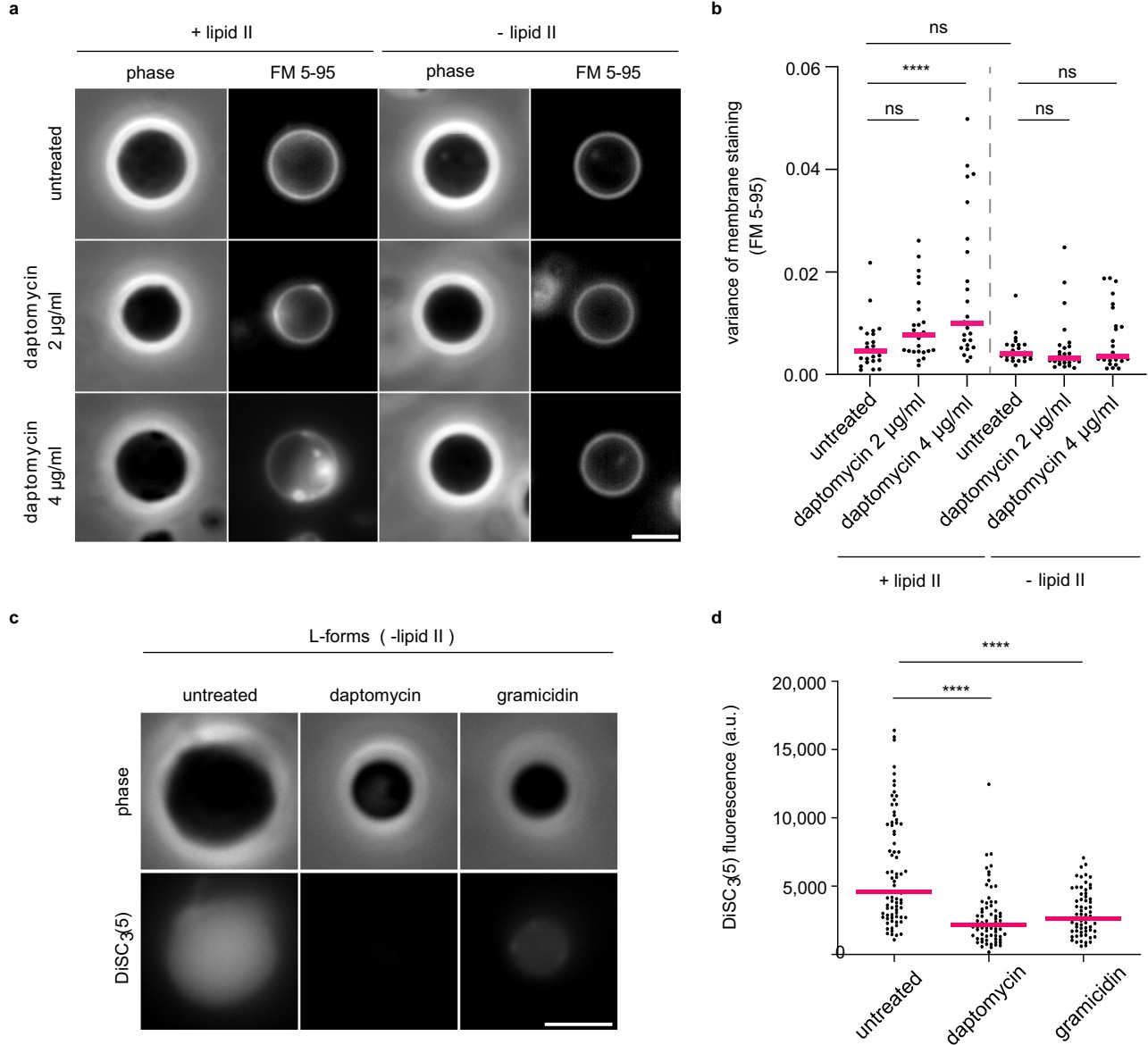

**Fig. 4 | Daptomycin requires interaction with undecaprenyl-coupled cell wall precursors for fluid membrane domain clustering, but not for membrane depolarisation. a** Phase contrast and fluorescence microscopy images of *B. subtilis* L-forms able and unable to produce lipid II stained with the membrane dye FM 5-95 and treated with either 2 μg/ml or 4 μg/ml of daptomycin. **b** Quantification of the variance of FM 5-95 for individual cells from the dataset shown in (**a**) (*n* = 25). Magenta lines indicate the median, while the *P* values represent the result of a one-way, unpaired ANOVA. **** represents *p* < 0.0001, ns, not significant. **c** Phase contrast and fluorescence microscopy of *B. subtilis* Δ*uppS* L-form cells stained with 1 μM

DiSC$_3$(5), in the absence and presence of 10 μg/ml daptomycin (25 min). The channel-forming peptide gramicidin was used as a positive control for membrane depolarisation (10 μM, 5 min). Scale bar, 3 μm. **d** Quantification of DiSC$_3$(5)-fluorescence for individual cells from the dataset shown in (**a**) (*n* = 65–78). Median fluorescence intensity is indicated with a magenta line, together with *P* values of a one-way, unpaired ANOVA. **** represents *p* < 0.0001. Strains used: *B. subtilis* AK092 (L-forms able to produce lipid II), *B. subtilis* AK0197 (L-forms unable to produce lipid II). For details about the sample size and statistical significance, see Supplementary Table 2.

and L-forms that lack cell wall precursor lipids altogether (Supplementary Figs. 3, and 4). We then performed FM 5-95 fluorescence microscopy, which allows the visualisation of daptomycin-induced fluid membrane regions[20], with both strains of *B. subtilis* L-forms. Whilst daptomycin retained the ability to induce fluid membrane domain clusters in L-forms producing cell wall precursor lipids, the cell membranes remained homogeneous in their absence (Fig. 4a, b). The ability of daptomycin to induce fluid membrane regions thus indeed relies on the presence of cell wall precursor lipids. Importantly, this phenomenon provides indirect in vivo support for the interaction between daptomycin and cell wall precursor lipids.

We next wanted to determine whether daptomycin-induced membrane depolarisation was also dependent on its interaction

with cell wall precursor lipids and the lipid domains associated with these complexes. Using the L-form strain lacking cell wall precursor lipids, we performed fluorescence microscopy of daptomycin-treated cells stained with DiSC$_3$(5), using the channel-forming peptide gramicidin as a positive control for membrane depolarisation[46]. As demonstrated by the microscopy images and quantification of single-cell fluorescence levels, the L-form cells exhibited DiSC$_3$(5) staining indicative of membrane potential, whilst daptomycin retained its ability to induce membrane depolarisation in this strain (Fig. 4c, d). These experiments demonstrate that, unlike the induction of lipid domains, the membrane depolarising activity of daptomycin is fully independent of its interaction with undecaprenyl-coupled cell wall precursors.

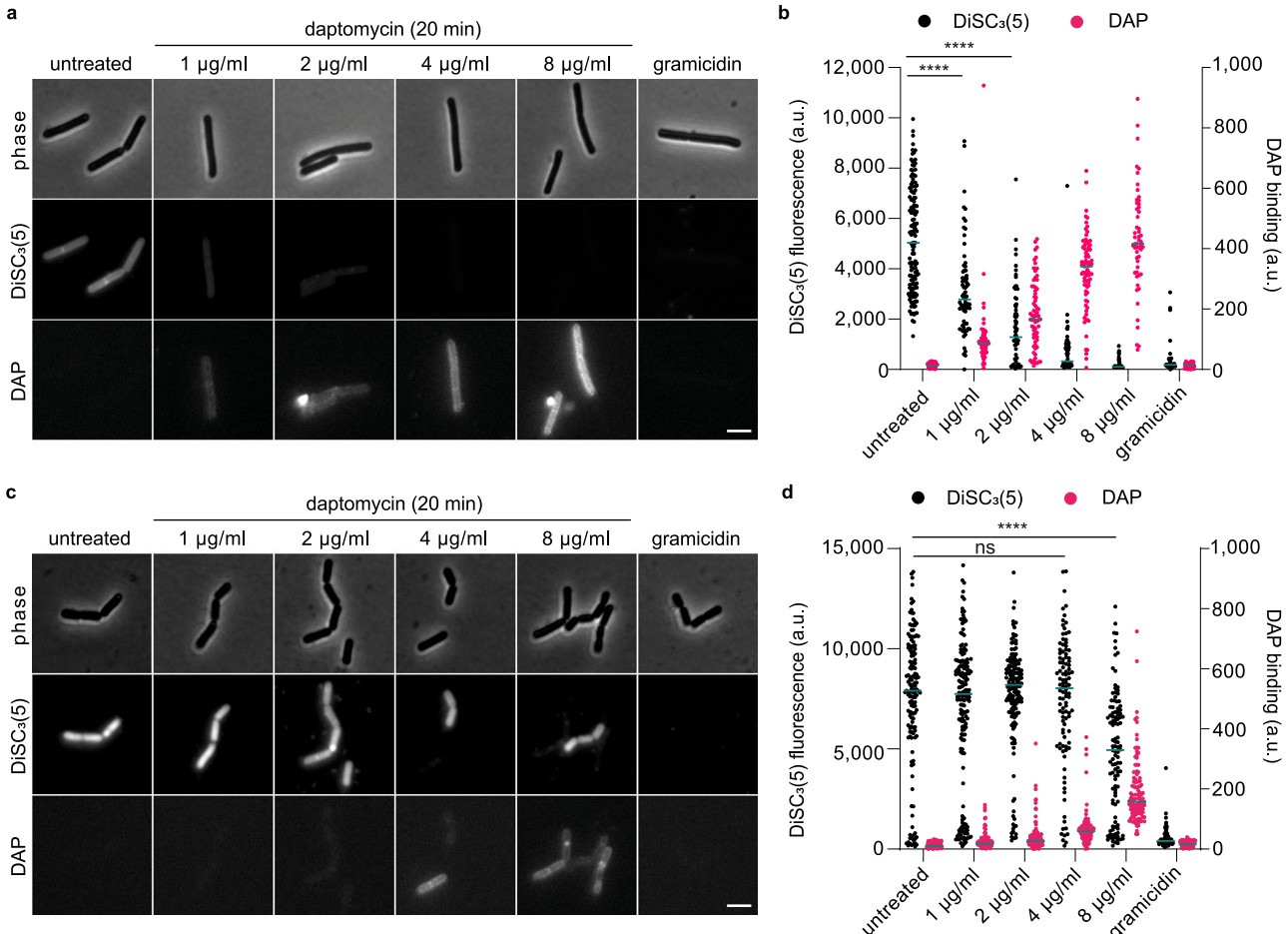

**Fig. 5 | *B. subtilis* membranes during the stationary growth phase bind daptomycin less effectively, therefore membrane depolarisation occurs at higher concentrations.** Phase contrast and fluorescence microscopy images of *B. subtilis* cells in (**a**) exponential and **c** stationary growth phase stained with 1 μM DiSC$_3$(5), untreated or treated with different daptomycin concentrations for 20 min. Daptomycin fluorescence was detected based on kynurenine autofluorescence (DAP). The channel-forming peptide gramicidin was used as a positive control for membrane depolarisation (10 μM, 5 min). Scale bar, 3 μm. **b**, **d** Quantification of DiSC$_3$(5)- and DAP-fluorescence for individual cells from the datasets shown in (**a**) and (**c**), respectively ($n = 40$–176). Median fluorescence intensity is indicated with a cyan line, together with $P$ values of a one-way, unpaired ANOVA. **** represents $p < 0.0001$, ns, not significant. Strain used: *B. subtilis* 168 (WT). For details about the sample size and statistical significance, see Supplementary Table 2.

The ability of daptomycin to depolarise cells in a cell wall precursor-independent manner suggests that daptomycin's antibacterial activity should not be limited to actively growing cells. To test this, we monitored daptomycin's capacity to bind stationary growth phase cells obtained from an overnight culture and correlated this binding with membrane potential measured through DiSC$_3$(5) staining. Binding to stationary growth phase cells was reduced (Fig. 5), likely due to the reduced membrane phosphatidylglycerol content[15,47]. However, increasing the daptomycin concentration to match previous binding levels resulted in clear membrane depolarisation (Fig. 5). Therefore, daptomycin's ability to depolarise membranes in vivo is indeed independent of the cells' growth stage.

In conclusion, daptomycin exhibits two independent mechanisms of action: (i) lipid domain formation that relies upon specific interaction with cell wall precursor lipids and is likely limited to actively growing cells, and (ii) membrane depolarisation triggered by interaction with regular membrane lipids, which does not depend on active growth.

### Large daptomycin-induced fluid membrane domains observed in *B. subtilis* depend on the IM30 homolog LiaH

Since daptomycin and other antibiotics that interfere with the lipid II biosynthesis cycle strongly induce the LiaRS two-component system[20,22,35], we wanted to investigate whether daptomycin's cell wall precursor interactions and the associated lipid domain formation are also modulated by the effector protein encoded by the *lia* regulon, LiaH. This possibility is further supported by the recent identification of LiaH as a member of the larger IM30/ESCRT-III membrane-remodelling superfamily[48–50]. For this aim, we performed fluorescence microscopy with the uncharged membrane fluorescent dye Nile Red, comparing wild-type cells with cells deficient for LiaH. In both strains, Nile red stained the membranes of unstressed cells in a uniform manner. Upon incubation with daptomycin, brightly stained Nile Red foci emerged in wild-type cells that co-localised with daptomycin clusters (Fig. 6, Supplementary Fig. 5, 6). In the absence of LiaH, however, these large domain clusters did not form, and instead daptomycin binding was concentrated on smaller, more frequent domains around the whole membrane, consistent with ongoing tripartite complex formation between daptomycin, lipid II and PG. The absence of lipid desaturase Des or lysyl-phosphatidylglycerol synthase MprF, both of which remodel membranes and have been implicated in daptomycin sensitivity[35], did not alter this lipid domain behaviour (Supplementary Fig. 7), thus arguing for a specific role of the Lia-system.

In contrast to *B. subtilis*, *S. aureus* does not encode homologs of LiaH or IM30 proteins in general. In this organism, a minor degree of

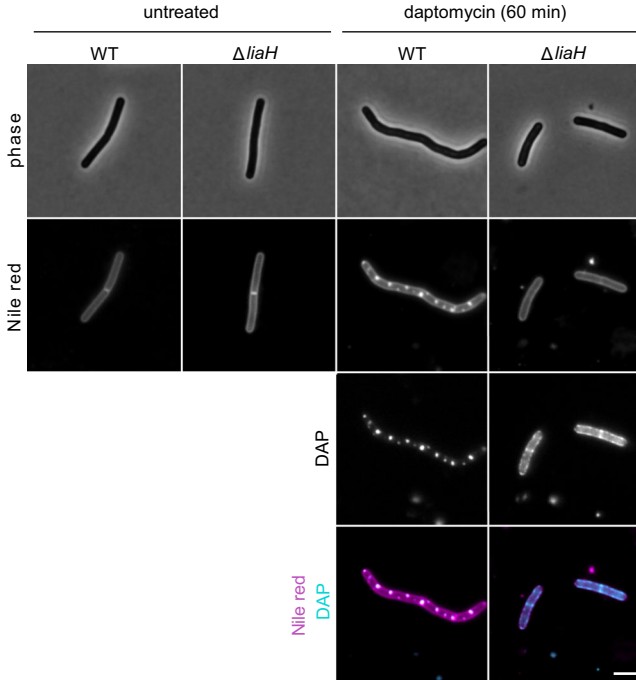

**Fig. 6 | LiaH plays a role in forming fluid daptomycin-lipid clusters.** Phase contrast and fluorescence microscopy of *B. subtilis* wild-type (WT) and Δ*liaH* cells stained with 1 μg/ml Nile red (shown in magenta) and treated with 4 μg/ml daptomycin (shown in cyan) for 60 min. Due to the intrinsic fluorescence of daptomycin, its localisation can be directly observed. Scale bar, 3 μm. Strains used: *B. subtilis* 168 (WT) and CD11 (Δ*liaH*).

lipid domain formation was observed following an extended (30–60 min) treatment with low concentrations of daptomycin (Fig. 7). The observed domains were, however, more subtle than the strong LiaH-dependent lipid domain clusters observed in *B. subtilis*, and potentially similar to domains observed in vitro upon binding to PG-containing liposomes[51]. Taken together, this implies that the *B. subtilis* IM30-homolog LiaH plays an intriguing role in clustering daptomycin-cell wall precursor lipid complexes. Further work is needed, and ongoing within our labs, to elucidate the interplay between daptomycin, cell wall precursor lipids and the IM30 protein LiaH.

## Discussion

The antibacterial mechanism of daptomycin has remained controversial due to disagreeing findings obtained from in vitro and in vivo experiments, and the difficulties of distinguishing primary effects from their secondary, more pleiotropic consequences in vivo[3,7,18,20,22,23,26–28,31]. Our data clarifies three aspects of daptomycin's complex antibacterial mechanism of action (Fig. 8). Firstly, by demonstrating daptomycin's ability to depolarise bacterial membranes independently of cell wall precursor lipids, our findings reconcile data obtained from in vitro model systems that argue for daptomycin-induced membrane conductivity for small cations[26,27] with the more recent discovery of cell wall precursor lipid interactions[22,23]. By showing that clustering of fluid membrane domains by daptomycin requires both cell wall precursor lipids and LiaH, this study also provides in vivo evidence for the interaction between daptomycin and cell wall precursor lipids, supporting the daptomycin binding data obtained in vitro[22]. Lastly, our observation that delayed membrane permeabilisation is a secondary cellular effect linked to autolysis explains why daptomycin-induced conductivity for large fluorophores can be observed in vivo, whilst in vitro conductivity is limited to small cations only[26,27].

The postulated dual mechanism of action for daptomycin is intriguing in the context of antimicrobial resistance (AMR). Daptomycin concentrations able to inhibit cell wall synthesis through interactions with cell wall precursor lipids, and those capable of depolarising the membranes lie very close to each other[20]. While this makes it difficult to evaluate the respective contributions of the two antibacterial mechanisms, it indicates that the cells face both stresses simultaneously, which likely suppresses the development of resistance. Moreover, non-growing cells remain sensitive to daptomycin's membrane-depolarising activity (Fig. 5), thus suppressing resistance development through mechanisms based on non-growing sub-populations. Finally, the ability to depolarise the membrane in a cell wall precursor-independent manner also explains why defence mechanisms, such as the Bce modules that protect Gram-positive bacteria from antimicrobial peptides that target cell wall precursor lipids[52–54], are not sufficient to protect from daptomycin. AMR towards daptomycin does develop, but it either employs species-specific mechanisms such as vesicle shedding combined with lack of phenol-soluble modulins in *S. aureus*[55] or is slow and/or multifactorial, as in the case of *B. subtilis*[35] and *E. faecium*[56]. The proposed dual mechanism of action, thus, provides a logical explanation for the difficulty to evolve high-level resistance towards daptomycin. However, as daptomycin's ability to interact with bacterial membranes and the specific interactions with cell wall precursor-lipids both rely on the phosphatidylglycerol (PG), it represents an overlap between the two otherwise largely independent mechanisms. Indeed, gain-of-function mutations in *mprF*, which catalyse the conversion of anionic PG to cationic lysyl-PG, frequently emerge in the clinic upon treatment of *S. aureus* infections with daptomycin[57–59]. While the exact molecular mechanisms through which these mutations confer daptomycin resistance are not fully understood, they are likely to facilitate resistance to both the cell wall precursor and depolarisation-linked antibacterial activities of daptomycin.

The recent prominent discoveries of teixobactin[60] and clovibactin[61], which both represent peptide antibiotics that target bacterial cell wall precursor lipids, have rekindled interest in this class of membrane-targeting antibiotics. What makes them attractive in terms of antibiotic development is their low resistance development potential, ability to overcome the resistance mechanisms currently in circulation, and improved selectivity towards bacterial membranes. Whilst positive in terms of AMR, identification and detailed study of such lead compounds is difficult due to the essential nature of the cell wall precursor lipids, necessitating the use of specialised in vitro approaches available in only few laboratories worldwide. The approach presented here, utilising L-forms, in which cell wall precursor lipids are not essential, provides a new approach for identifying novel cell wall precursor-targeting antibiotics, and a promising route towards studying their detailed mechanism of action in vivo.

## Methods

### Strains, media and growth conditions

Strains and genotypes are listed in Table 1. *S. aureus* and *B. subtilis* strains were aerobically grown at 37 °C in lysogeny broth (LB) (Miller) (10 g/l tryptone, 5 g/l yeast extract, 10 g/l NaCl) supplemented with 1.25 mM CaCl$_2$, with the exception of *B. subtilis* L-forms which were grown at 30 °C in a specialised medium detailed below.

### Generation and growth inhibition of *B. subtilis* L-forms

For generation of *B. subtilis* L-forms, strains were grown in LB with any appropriate antibiotics or inducers to mid-logarithmic phase at 37 °C with shaking. Cells were then washed in fresh LB and resuspended in Nutrient Broth (NB) (Oxoid) (1 g/l 'Lab-Lemco' powder, 2 g/l yeast extract, 5 g/l peptone, 5 g/l NaCl) supplemented with MSM (20 mM maleic acid, 20 mM MgCl$_2$, 0.5 M sucrose) and 2 mg/ml lysozyme for an hour with gentle shaking to convert rod-shaped cells

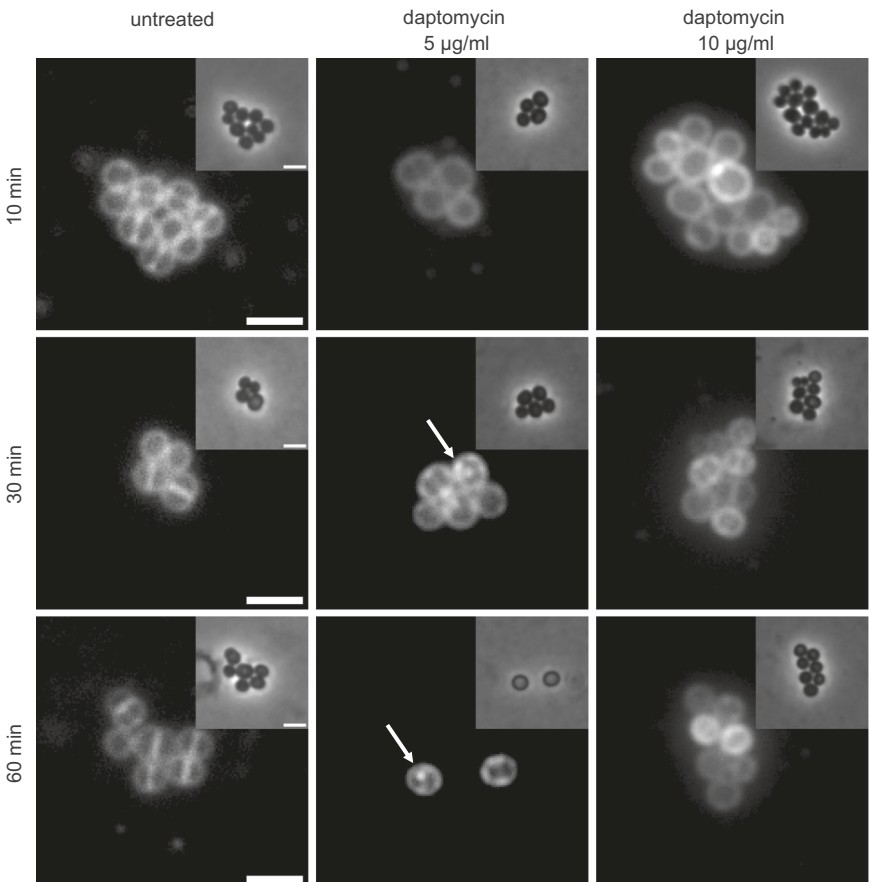

untreated | daptomycin 5 µg/ml | daptomycin 10 µg/ml

**Fig. 7 | Daptomycin induces subtle fluid membrane domains in *S. aureus*.** Phase contrast (insert) and fluorescence microscopy images of *S. aureus* cells stained with 1 µg/ml FM 4-64 and treated with either 5 µg/ml or 10 µg/ml daptomycin. Scale bar, 2 µm. White arrows indicate daptomycin-induced fluid lipid clustering. Note, here the membrane dye FM 4-64 was used as it produced better staining in *S. aureus*. Strain used: *S. aureus* SH1000.

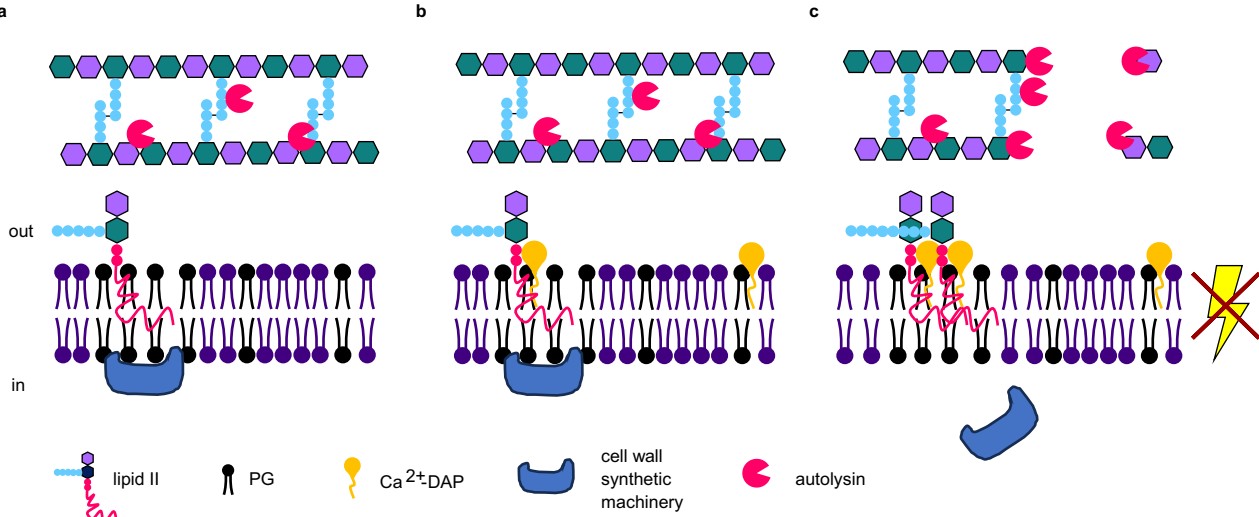

lipid II | PG | Ca²⁺-DAP | cell wall synthetic machinery | autolysin

**Fig. 8 | Proposed model of daptomycin dual mechanisms of action. a** Peripheral cell wall synthetic proteins and the undecaprenyl-coupled peptidoglycan precursor lipid II localise to fluid microdomains under normal growth conditions. **b** Calcium-daptomycin oligomers (Ca²⁺-DAP) form a tripartite complex with phosphatidylglycerol (PG) and lipid II or bind directly to anionic phospholipids such as PG. **c** Tripartite complex formation inhibits cell wall synthesis by steric hindrance and by inducing detachment of cell wall synthetic proteins. The direct (lipid II-independent) interaction between Ca²⁺-DAP and anionic phospholipids triggers membrane depolarisation and associated autolysis.

## Table 1 | Strains used in this study

| Strain name | Genotype | Reference |
|---|---|---|
| *B. subtilis* 168 | *trpC2* | 66 |
| *B. subtilis* KS19 | *lytABC::neo lytD::tet lytE::cat lytF::spc* | 67 |
| *B. subtilis* AK092 | *ispA xscb:TmKm (kan) amyE::spc Pxyl-accDA* | This work. |
| *B. subtilis* AK0197 | *ispA(D92E)::spc uppS::kan pLOSS (Pspac-uppS) (mls)* | This work. |
| *B. subtilis* BKK33120 | *liaH::kan* | 63 |
| *B. subtilis* CD11 | *trp+ ΔliaH* | This work. |
| *B. subtilis* BKK08425 | *mprF::kan* | 63 |
| *B. subtilis* BKK19180 | *des::kan* | 63 |
| *B. subtilis* KS50 | *des::kan* | This work. |
| *B. subtilis* AK0118 | *mprF::kan* | This work. |
| *B. subtilis* RM81 | *ispA xscb:TmKm (kan)* | 45 |
| *B. subtilis* YK1738 | *amyE::spc Pxyl-accDA* | 45 |
| *B. subtilis* HS34 | *mreD:pMutin4 (mreD1-157) amyE::spc Pxyl-gfp-mreB* | 62 |
| *S. aureus* SH1000 | *8325-4 rsbU+* | 68 |

Gene and species names in italics.

## Table 2 | Oligonucleotides

| Oligonucleotide | Sequence |
|---|---|
| oAK227 | GTAGAAAGGAGATTCCTAGGATGCTCAACATACTCAAAAATTGGAAG |
| oAK282 | TACCGTTCGTATAGCATACATTATAC |
| oAK283 | TCTACCGTTCGTATAATGTATGCTATAC |
| oAK288 | AAGTTGCTGAACCGTACATAAGAAG |
| oAK289 | GTATGCTATACGAACGGTAGAGATTCCTCCGTCACCC |
| oAK290 | ACATTATACGAACGGTAGATTTCAGCAGAGAGGCCGGAG |
| oAK291 | GTATCCAGATCACCGACAGAGAC |
| oAK320 | GAAGGATGGCGTGCTGACCTGTG |
| oAK374 | CAAATCGTCTTCATCCATGCATGG |
| oAK375 | CATGCATGGATGAAGACGATTTGC |
| oAK376 | TTAGTGATCTCTTGCCGCAATTAAATC |
| oAK377 | GCGGCAAGAGATCACTAAGCTCATCCATGCGACTCTAG |
| oAK378 | GCTTCAAATTAGGATCGTTCTAATTGAGAGAAGTTTCTATAGAATTTTTC |
| oAK379 | AACGATCCTAATTTGAAGCGGC |
| oAK380 | CGTATTGAAGATTCTCAATCAGCTC |
| oAK386 | CCAGCTTGTTGATACACTAATGC |
| oAK387 | TAATTGTGAGCGCTCACAATTC |
| oAK388 | GAATTGTGAGCGCTCACAATTATAGAAAGGAGATTCCTAGGATG |
| oAK396 | TAGTGTATCAACAAGCTGGCTAAATTCCGCCAAACCTCCGG |

Custom oligonucleotides synthesised by Eurogentec.

to protoplasts. L-forms were then propagated by diluting the protoplasts in NB/MSM supplemented with penicillin G (200 μg/ml) and growing at 30 °C without shaking for 2–4 days. For growth inhibition experiments, L-forms were diluted to an $OD_{600}$ of 0.1 and left untreated or treated with 10 μg/ml daptomycin. Cultures were then incubated at 30 °C without shaking until visible growth was observed.

### Strain construction
To clone the *ispA(D92E)* mutation, four DNA fragments were amplified using specific primers (Table 2): the first fragment was amplified from 168 gDNA using the primer pair oAK320 and oAK374. This fragment included the upstream region of *ispA*, along with the 3′ end of the *xseA*

gene, while incorporating the *ispA(D92E)* mutation at the 3′ end. A second fragment was amplified from 168 gDNA using the primers oAK375 and oAK376. This fragment introduced the *ispA(D92E)* mutation at its 5′ end, and at the 3′ end, it contained a sequence complementary to the spectinomycin resistance marker. The third fragment was amplified using primers oAK377 and oAK378 to isolate the spectinomycin resistance marker from HS34 gDNA, while also adding a sequence complementary to the *dxs* gene at its 3′ end. The fourth fragment was amplified from 168 gDNA using primers oAK379 and oAK380, targeting the *dxs* gene. The linear PCR products were gel-purified and assembled into a single construct using NEBuilder HiFi DNA Assembly before transformation. To construct the Δ*uppS* deletion, three DNA fragments were amplified using specific primers. The first fragment was amplified from 168 gDNA using the primer pair oAK288 and oAK289. This fragment included the 3′ end of the *frr* gene and extended up to the region immediately preceding the *uppS* start codon. A second fragment was generated to introduce a kanamycin resistance cassette. This was amplified using primers oAK282 and oAK283, which added complementary sequences at both the 3′ and 5′ ends of the cassette for efficient recombination. The kanamycin resistance cassette was derived from ref. 58. The third fragment, containing the *cdsA* and *dxR* sequences, was amplified from 168 gDNA using primers oAK290 and oAK291. Each linear PCR product was gel-purified, and the fragments were assembled into a single construct using NEBuilder HiFi DNA Assembly before transformation. To construct the pLOSS-*uppS* plasmid, the *uppS* coding sequence was amplified from 168 gDNA using primer pair oAK227 and oAK396. Primer oAK227 introduced a ribosome binding site previously used for expressing msfGFP-MreB[62] directly upstream of the *uppS* start codon. Primer oAK396 adds a complementary sequence at the 3′ end for efficient cloning into the pLOSS vector. A second round of PCR was performed using primers oAK388 and oAK396 to introduce pLOSS-specific complementary sequences at the 5′ end of the amplified product, facilitating seamless insertion into the pLOSS vector. The pLOSS plasmid backbone was amplified separately using primers oAK387 and oAK386. After gel purification of all linear PCR products, the *uppS* gene fragment was assembled into the pLOSS plasmid using NEBuilder HiFi DNA Assembly before transformation. Ultimately, *ispA(D92E)*, Δ*uppS* and pLOSS-*uppS* were combined, resulting in strain AK0197. To generate the strain AK092, *B. subtilis* 168 was transformed with genomic DNA from strains RM81 and YK1738. To generate the strain CD11, *B. subtilis* MDS23 (168 *trp*+) was transformed with genomic DNA from BKK33120 (*liaH::kan*), followed by removal of the resistance cassette as described in Koo et al.[63]. To generate the strains KS50 and AK0118, *B. subtilis* 168 was transformed with genomic DNA from BKK08425 (*mprF::kan*) and BKK19180 (*des::kan*), respectively.

### Growth experiments
Overnight cultures were diluted to an $OD_{600}$ of 0.05 and transferred to a 96-well plate at a final volume of 100 μL per well. Cells were then incubated in a BMG CLARIOstar plate reader under constant agitation. After an $OD_{600}$ of 0.3–0.4 was reached, antibiotics were added, and $OD_{600}$ measurements were recorded for up to 10 h.

### Used antibiotic and dyes
Daptomycin was purchased from Abcam; nisin, gramicidin ABCD, and Nile red from Sigma-Aldrich; SYTOX Green, FM 5-95, BODIPY-FL vancomycin, and FM 4-64 from Thermo Fisher Scientific; and $DiSC_3(5)$ from Anaspec. Daptomycin, nisin, FM 5-95, and SYTOX Green were dissolved in sterile water. Gramicidin ABCD, Nile red, $DiSC_3(5)$, BODIPY-FL vancomycin, and FM 4-64 were dissolved in sterile DMSO.

### Minimum inhibitory concentration (MIC) assay
Overnight cultures were diluted 1:100 in appropriate growth medium and grown to mid-logarithmic phase. Cells were then diluted to give a

final concentration of $5 \times 10^5$ cells per well in a prewarmed 96-well microtitre plate. This plate was prepared with an initial high concentration of the desired compound followed by a serial twofold dilution. After addition of the cells, the plate was incubated at 37 °C for 16 h with shaking at 700 rpm. The MIC was defined as the lowest compound concentration able to inhibit visible bacterial growth.

## Fluorometric determination of membrane pore formation
Cultures were grown to logarithmic growth phase and, if needed, adjusted to an $OD_{600}$ of 0.5. SYTOX Green was then added to cells at a final concentration of 1 μM and cells were transferred to black polystyrene 96-well plates (Porvair Sciences). Fluorescence intensity was monitored until a stable baseline was obtained, followed by the addition of 10 μg/ml daptomycin. Measurements were taken every minute for 150 min using a BMG CLARIOstar multimode plate reader equipped with 485 nm (±10) excitation, and 520 nm (±10) emission filters. All media, plates, and instruments were warmed to 37 °C prior to use.

## Fluorescence microscopy
Samples (0.5 μl) were immobilised on Teflon-coated multi-spot microscope slides (Thermo Fisher) covered with a thin layer of 1.2% agarose (in $H_2O$) and imaged immediately. Fluorescence microscopy of *B. subtilis* was performed using a Nikon Eclipse Ti equipped with a Nikon Plan Apo 100×/1.40 Oil Ph3 objective, CoolLED pE-4000 light source, Photometrics BSI sCMOS camera, and the filter sets: Chroma 49000 (EX350/50, DM400lp, EM460/50) used for daptomycin, Chroma 49002 (EX 470/40, DM495lpxr, EM 525/50) used for SYTOX Green and BODIPY FL-vancomycin, Chroma 49008 (EX560/40, DM585lprx, EM630/75) used for FM 5-95, and Nile red, and Semrock Cy5-4040C (EX 628/40, DM660lp, EM 692/40) used for $DiSC_3(5)$; while *S. aureus* samples were imaged using a Nikon Eclipse Ti2 equipped with a Photometrics PRIME BSI EXP camera, Lumencor Sola SE II FISH 365 light source and CFI Plan Apochromat DM Lambda 100X Oil objective (N.A. 1.45, W.D. 0.13 mm). The filter used for $DiSC_3(5)$ was the same as above; whilst Semrock GFP-4050B was used for SYTOX Green, and Semrock NKBV-0014 for FM 4-64. All images were obtained using the Nikon NIS elements AR software version 5.42.02 and processed and analysed with Fiji[64].

For fluorescence microscopy of *B. subtilis* L-forms, samples (70 μl) were transferred to a microscope slide containing a gene frame (1.7 cm × 2.8 cm; Thermo Fisher Scientific) and sealed with a plasma-treated coverslip (20 min plasma treatment in a plasma cleaner; Harrick Plasma). The sealed slide was then inverted and allowed to stand for 15–20 min to allow L-forms to settle, and images were acquired using the microscope and camera setup stated above for *B. subtilis*. For fluorescent vancomycin staining, BODIPY-FL vancomycin was mixed one to one with unlabelled vancomycin and added to L-form cells at a final concentration of 1 μg/ml for 30 min. L-forms were then imaged as earlier described.

## Image analysis
Quantification of $DiSC_3(5)$ and SYTOX Green-fluorescence for individual cells was performed using Fiji[64]. In brief, any background fluorescence was first subtracted, originating from unincorporated dye and medium. For *B. subtilis*, individual cells were identified and converted to regions of interest by thresholding of corresponding phase contrast images. If cells adhered to each other or grew as chains, they were separated manually prior to automated cell detection. For *S. aureus*, segmentation was performed utilising the Weka segmentation plugin[65]. Cells that were not correctly detected by this tool were manually segmented, and out of focus cells were discarded from the analysis. Finally, mean fluorescence intensity values for individual cells across the population were obtained.

## Statistics and reproducibility
All experiments were conducted with at least two independent biological replicates, yielding similar results. Most experiments were carried out as biological triplicates or more. A representative dataset is presented in the figures and graphs. Statistical significance was analysed using one-way, unpaired ANOVA. For details about sample sizes and *p*-values, see Supplementary Table 2.

## Reporting summary
Further information on research design is available in the Nature Portfolio Reporting Summary linked to this article.

## Data availability
Source data for all figures and graphs presented in the manuscript are available via Newcastle University's research data repository at https://doi.org/10.25405/data.ncl.29656607.

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

## Acknowledgements

Funding for this research was provided by a Medical Research Council (MRC) DTP Studentships MR/N013840/1 (J.A.B) and MR/W006944/1 (J.W), UKRI Future Leaders Fellowship MR/W009587/1 (K.M), and Biotechnology and Biological Sciences Research Council (BBSRC) grants BB/S00257X/1 and BB/X003035/1 (H.S). We would like to thank James Grimshaw for technical assistance with image analysis, Martin Andersson (Chalmers University of Technology) for access to BSL-II facilities, Kenneth H. Seistrup and Lauren Elliott for strains and preliminary experiments leading to this project, Chris Duringer for construction of strains, and Susanne Gebhard and Leendert Hamoen for providing feedback to the manuscript during its preparation.

## Author contributions

J.A.B., A-B.S., A.K., and J.W. performed the experiments, J.A.B., A.K., K.M., M.W., and H.S. designed experiments and helped with data interpretation. H.S. and M.W. conceived and supervised the project. J.A.B. and H.S. wrote the manuscript, with help from A.K., M.W., A-B.S., and K.M.

## Competing interests

The authors declare no competing interests.
