## [Transparent Peer Review file · Nature Communications]

The last resort antibiotic daptomycin exhibits two independent antibacterial mechanisms of action

Corresponding Author: Professor Henrik Strahl

Version 0:

Reviewer comments:

Reviewer #1

(Remarks to the Author)

Daptomycin is a highly efficient last-resort antibiotic that targets the bacterial cell membrane. Despite its clinical importance, the exact mechanism by which daptomycin kills bacteria is not fully understood. Different experiments have led to different models, including (i) blockage of cell wall synthesis, (ii) membrane pore formation, and (iii) the generation of altered membrane curvature leading to aberrant recruitment of proteins. Thus, the explicit mode of daptomycin action has remained controversial. In their study, the authors find that daptomycin exhibits two distinct and independent mechanisms of action: i) interaction with cell wall precursor lipids resulting in clustering of lipid II and associated fluid membrane microdomains and (ii) depolarisation of the cytoplasmic membrane that is independent of interactions with cell wall precursor lipids. In addition, the experiments further provide, for the first time, evidence of a specific role of the LiaH protein in cell membrane organization.

The experiments conducted in this study are sound, concise and consistent. The manuscript is well written and gives new and important insights into the mechanism of action of this important antibiotic, delivering a plausible explanation for the often contradictory results observed in previous *in vivo* and *in vitro* studies.

Minor comments

L36ff

The differentiation “mechanism of action (direct drug-target interactions leading to inhibition) and mode of action (cellular consequences of target inhibition)” seems awkward. I suggest rephrasing as follows: “mechanism of action, both in terms of direct drug-target interactions leading to inhibition and cellular consequences of target inhibition.”

L60

daptomycin-treated *B. subtilis*

Figure 1ab

Provide figure legend (Nis, DAP, untreated) for each 1a and 1b individually

Figure 1e

figure legend, add for clarity: kynurenine autofluorescence (DAP)

L135, 148, 211, 213, 567, 569

wild-type cells

Fig 2 may be moved to the SI section, or integrated in Fig 1 (plus SI).

L185f

fluid membrane regions instead of lipid domains

L231

"Our data clarifies" instead of "We believe this manuscript brings clarity to"

Figure 7

Ca²⁺-dap vs. Ca²⁺-DAP

Reviewer #2

(Remarks to the Author)

I find the submitted publication potentially interesting, with important conclusions. The topic is relevant, essential for the scientific community.

However, I must state that the work presents a large number of microscopic images, and some of them are inadmissibly manipulated by some of the authors. For this reason, I cannot recommend the work for publication. I will therefore not discuss the remaining scientific comments in detail here (e.g. too low a number of examined cells; Fig. 1d, Fig. 2b).

Below I attach an analysis of my procedure and what I found as obvious problems in the publication.

I was interested in the fact that in Fig. 1e there is a low intensity in one surviving cell, specifically the combination of 10 mg/l DAP, DAP channel. This forced me to extract the original microscopic images from the pdf file. I used the pdfimages program (<https://www.systutorials.com/docs/linux/man/1-pdfimages/>). For example, the command:

```
pdfimages -tiff -f 34 -l 34 582625_0_merged_1738595520.pdf ~/p34
```

(Note: The images I attach have been intentionally significantly increased in brightness to make the details clear.)

Figure 1e (10 mg/l DAP) has been adjusted in terms of brightness and contrast (i.e. the range of values displayed). While the DAP channel is probably in its original quality, the SYTOX channel is on a completely dark background (value 0 on an 8-bit scale). It is not clear why DAP exhibits different localization on individual cells (membrane and cytosolic). There is a risk of overlapping fluorescence channels.

Figure 1e (10E-5 M nisin) shows completely different cells in the DAP channel than in the other two channels!

Figure 1e shows a completely black image in the nisin-DAP channel (intensity 0 on all pixels), as does the daptomycin-SYTOX channel. This is not real recorded data.

Figure 4c shows three different cells, but at different magnifications. The difference is 1.3x. The difference in cell size is ~2x. The contrast on the gramicidin cell was additionally adjusted (possibly gamma value).

It is possible that these are just some errors on the part of the person who prepared the images. Given the above, the work does not seem sufficiently credible and I do not recommend it for publication.

Reviewer #3

(Remarks to the Author)

Buttress et al., describe efforts to unravel the mechanism(s) by which daptomycin acts on gram-positive bacteria. As the authors rightly state, this is an area that has been subject to much debate and this work helps to provide some clarity. As such, the work is valuable and advances our understanding. However, the conclusions drawn and insight provided would be much stronger with some additional data, controls and analyses.

Major points

The authors do not attempt to understand the relative contributions of each mechanism of action to overall susceptibility. Is one mechanism more important than the other? Similarly, in terms of resistance mechanisms, do e.g. gain of function *mprF* mutations in *S. aureus* affect both mechanisms or just one? Additionally, the role of *mprF* in conferring resistance should be discussed in more detail.

Does growth phase affect the experimental outcomes? Cell wall inhibitors typically have much larger effects in log phase cultures, relative to stationary phase, whereas membrane disruption is typically less affected.

The daptomycin susceptibility of strains (MIC) should be provided to place findings in context. Previous work from this group showed dose-dependent effects of daptomycin on the inhibition of cell wall synthesis and membrane damage leading to lysis [reference 20 in the manuscript]. As such, the relationship between antibiotic concentration used and MIC needs to be made clear, especially since the authors seek to compare two different bacterial species.

The *liaH*-deficient mutant does not form daptomycin-lipid clusters, which the authors propose contributes to the antibacterial activity of the antibiotic. This would suggest that the *liaH*-mutant is less susceptible to daptomycin. However, previous work has shown that a *liaH* mutant is more susceptible to daptomycin - how do the authors explain this?

As an additional point for the role of LiaH, which is an interesting finding, the absence of complementation or comparison with other mutants reduces the strength of this finding.

The authors provide evidence that the signal from sytox green is due to lysis triggered by daptomycin. However, a similar analysis is missing for *S. aureus*, which does not appear to lyse in the presence of daptomycin (supp figure 1). Instead, the authors employ assays to measure membrane depolarisation. As such, it is difficult to directly compare the effects of daptomycin on *B. subtilis* and *S. aureus*.

Minor Points

Lines 30-31: please make clear that daptomycin has increased affinity relative to other bacterial phospholipids, otherwise 'increased affinity' lacks a comparator.

Line 59: please remove 'rather' as it reduces clarity as currently written.

Line 76: Please replace 'at last' with 'finally'.

Line 170. It should be mentioned that another study failed to replicate this finding (Kotsogianni et al., 2021). Whilst this may reflect different experimental approaches, it would help to make the authors' point that the role of lipid II was worth exploring further.

Lines 187-188: I would be cautious here since the studies do not measure direct binding. It may be that the presence of lipid II has indirect effects via e.g. membrane fluidity.

Line 193: why use gramicidin and not nisin here, as used earlier in the manuscript?

Version 1:

Reviewer comments:

Reviewer #3

(Remarks to the Author)

The authors have fully addressed my concerns.

REPLY TO REVIEWER COMMENTS

Reviewer #1:

Daptomycin is a highly efficient last-resort antibiotic that targets the bacterial cell membrane. Despite its clinical importance, the exact mechanism by which daptomycin kills bacteria is not fully understood. Different experiments have led to different models, including (i) blockage of cell wall synthesis, (ii) membrane pore formation, and (iii) the generation of altered membrane curvature leading to aberrant recruitment of proteins. Thus, the explicit mode of daptomycin action has remained controversial. In their study, the authors find that daptomycin exhibits two distinct and independent mechanisms of action: i) interaction with cell wall precursor lipids resulting in clustering of lipid II and associated fluid membrane microdomains and (ii) depolarisation of the cytoplasmic membrane that is independent of interactions with cell wall precursor lipids. In addition, the experiments further provide, for the first time, evidence of a specific role of the LiaH protein in cell membrane organization. The experiments conducted in this study are sound, concise and consistent. The manuscript is well written and gives new and important insights into the mechanism of action of this important antibiotic, delivering a plausible explanation for the often contradictory results observed in previous in vivo and in vitro studies.

We thank the reviewer for the positive comments.

Minor comments

L36ff

The differentiation “mechanism of action (direct drug-target interactions leading to inhibition) and mode of action (cellular consequences of target inhibition)” seems awkward. I suggest rephrasing as follows: “mechanism of action, both in terms of direct drug-target interactions leading to inhibition and cellular consequences of target inhibition.”

As suggested by the reviewer, we have now rephrased to “mechanism of action, both in terms of direct drug-target interactions leading to inhibition and cellular consequences of target inhibition (frequently termed mode of action)”.

New lines 37-38.

L60

daptomycin-treated *B. subtilis*

We have amended this in the text as suggested.

New line 61.

Figure 1ab

Provide figure legend (Nis, DAP, untreated) for each 1a and 1b individually

We have amended the figure and the legend as suggested.

Figure 1e

figure legend, add for clarity: kynurenine autofluorescence (DAP)

We have added this to the figure legend as recommended.

L135, 148, 211, 213, 567, 569

wild-type cells

We have made suggested changes to this throughout the text.

Fig 2 may be moved to the SI section, or integrated in Fig 1 (plus SI).

In our view, differentiating between the two routes for membrane permeabilisation (direct membrane activity and indirect permeabilisation through induced autolysis) is important for our conclusion. Moreover, the role of autolysis is frequently overlooked when studying the activity of membrane antibiotics *in vivo*. The approach we have used is, thus, informative for the community. Given that the space is available, we prefer to keep the data in the main manuscript.

L185f

fluid membrane regions instead of lipid domains

As the reviewer suggests, we have changed “lipid domains” to “fluid membrane regions”.

New line 189.

L231

“Our data clarifies” instead of “We believe this manuscript brings clarity to”

As suggested, we have replaced “We believe this manuscript brings clarity to” with “Our data clarifies”.

New line 248.

Figure 7

Ca²⁺-dap vs. Ca²⁺-DAP

We thank the reviewer for spotting this. Ca²⁺-dap has been amended to Ca²⁺-DAP throughout the figure legends.

Reviewer #2:

I find the submitted publication potentially interesting, with important conclusions. The topic is relevant, essential for the scientific community.

We thank the reviewer for sharing a positive view of the relevance of this manuscript's topic.

However, I must state that the work presents a large number of microscopic images, and some of them are inadmissibly manipulated by some of the authors. For this reason, I cannot recommend the work for publication. I will therefore not discuss the remaining scientific comments in detail here (e.g. too low a number of examined cells; Fig. 1d, Fig. 2b).

We sincerely apologise for the inadvertent mistakes made during the preparation of some of the figures and provide explanations for the concerns raised by the reviewer in the point-to-point responses below. We addressed all the concerns in the revised version of the manuscript and, for full transparency, provide all the raw data, including microscopy images in the raw format as captured upon imaging at <https://doi.org/10.25405/data.ncl.29656607>. While we acknowledge that sufficient care was not taken in preparing the pre-publication quality figures, we would like to stress that this had no impact on the conclusions drawn from the imaging data.

We have increased the number of examined cells for Fig. 1d and Fig. 2b as suggested.

Below I attach an analysis of my procedure and what I found as obvious problems in the publication.

I was interested in the fact that in Fig. 1e there is a low intensity in one surviving cell, specifically the combination of 10 mg/l DAP, DAP channel. This forced me to extract the original microscopic images from the pdf file. I used the pdfimages program (<https://www.systutorials.com/docs/linux/man/1-pdfimages/>). For example, the command:

```
pdfimages -tiff -f 34 -l 34 582625_0_merged_1738595520.pdf ~/p34
```

(Note: The images I attach have been intentionally significantly increased in brightness to make the

details clear.)

Figure 1e (10 mg/l DAP) has been adjusted in terms of brightness and contrast (i.e. the range of values displayed). While the DAP channel is probably in its original quality, the SYTOX channel is on a completely dark background (value 0 on an 8-bit scale). It is not clear why DAP exhibits different localization on individual cells (membrane and cytosolic). There is a risk of overlapping fluorescence channels.

The wide dynamic range of SYTOX Green fluorescence necessitates imaging samples with low excitation light and short exposure times. Consequently, the background fluorescence, consisting of medium background fluorescence and camera read noise, is very low while the SYTOX signal remain very high. When these 16-bit raw captured images are converted to 8-bit images for web and print, the very low background typically results in zero values unless the zero-intensity values in both image types are intentionally matched. However, such conversion would cause the real signal of interest (high SYTOX fluorescence caused by DNA interaction) to appear saturated. We maintain that converting between 16-bit and 8-bit images in this manner complies with the NCOMMs image preparation guidelines and the community standards. The associated 16-bit raw data is now provided to remove any ambiguity related to our image processing.

The cytosolic versus membrane localisation of daptomycin-fluorescence is caused by the difference in the degree of lysis between these cells, rather than spectral bleed-through. In fact, comparing the signals directly makes it abundantly clear that pixels featuring very high SYTOX signals (Fig 1E, 10 µg/ml daptomycin, SYTOX Green) do not feature high signals in the DAP-channel, thus ruling out significant spectral bleed-through.

Figure 1e (10E-5 M nisin) shows completely different cells in the DAP channel than in the other two channels!

This is indeed a mistake that was made during the preparation of the figures, resulting in the incorrect (visually black) image being used. We sincerely apologise for this mistake and thank the reviewer for pointing it out. This has now been corrected, and the raw data is made available to eliminate any remaining ambiguity. We would like to stress, however, that this does not alter the conclusion, as these images feature no fluorescence signal that is interpreted (they are essentially black).

Figure 1e shows a completely black image in the nisin-DAP channel (intensity 0 on all pixels), as does the daptomycin-SYTOX channel. This is not real recorded data.

Please see above for the explanation regarding the conversion from 16-bit data (as captured by the microscope camera) to 8-bit format, which is required for visualisation on computer screens or printing. There is no technical way to visualise or print the original recorded (16-bit) data without converting it to 8-bit. This conversion is a standard process in microscopy that must be carried out with care to avoid misrepresenting the original 16-bit data, but that cannot be avoided altogether. The associated 16-bit raw data is now provided to prevent any remaining ambiguity.

Figure 4c shows three different cells, but at different magnifications. The difference is 1.3x. The difference in cell size is ~2x. The contrast on the gramicidin cell was additionally adjusted (possibly gamma value).

We thank the reviewer for spotting this mistake. One of the images was indeed in a different scale. This has now been corrected. In general, a high degree of variability in cell size is typical for *B. subtilis* L-form cultures. Concerning the histograms of the DISC images, they are as expected from the 16 to 8-bit conversion when applied using the same contrast settings for the different images (which is needed to preserve the relative apparent fluorescence intensities). Concerning the gramicidin image, a conversion error has indeed occurred that is difficult to trace back, resulting in the odd-looking histogram. This has now been corrected. Again, we would like to stress that this does not alter the conclusion drawn from the images (which is based on the quantification of a larger number of cells

rather than the appearance of a single cell provided in the same figure), and the associated 16-bit raw data is now provided to eliminate any remaining ambiguity.

It is possible that these are just some errors on the part of the person who prepared the images. Given the above, the work does not seem sufficiently credible and I do not recommend it for publication.

We hope that we have been able to convince the reviewer that while a few of the issues highlighted were indeed honest errors from our side, they do not alter any conclusion drawn from the data (only affect fields that have low-no signal). We hope that the full transparency with the provided raw data removes the remaining concerns.

Reviewer #3:

Buttress et al., describe efforts to unravel the mechanism(s) by which daptomycin acts on gram-positive bacteria. As the authors rightly state, this is an area that has been subject to much debate and this work helps to provide some clarity. As such, the work is valuable and advances our understanding. However, the conclusions drawn and insight provided would be much stronger with some additional data, controls and analyses.

We thank the reviewer for the positive comments. We have now added additional data and controls discussed in the point-to-point response below.

Major points

The authors do not attempt to understand the relative contributions of each mechanism of action to overall susceptibility. Is one mechanism more important than the other?

This is a very intriguing question, but one that is experimentally difficult to answer. Currently, the only experimental *in vivo* method involves using L-form cells (thus removing the influence of cell wall synthesis inhibition) but, due to the different media required and the numerous physiological changes the cells undergo during adaptation to growth without a cell wall, we are cautious about drawing strong conclusions from minor differences in growth-inhibitory concentration and activities between the two cell types and media. We have now included additional discussion (lines 260-266) referring to this valid question.

Similarly, in terms of resistance mechanisms, do e.g. gain of function *mprF* mutations in *S. aureus* affect both mechanisms or just one? Additionally, the role of *mprF* in conferring resistance should be discussed in more detail.

As both mechanisms require PG, it is indeed likely that *S. aureus mprF* gain-of-function mutations could provide resistance to both mechanisms. However, testing these mutations in *S. aureus* would require a well-controlled *S. aureus* L-form model that does not rely on other cell wall antibiotics to induce the L-form state (as is currently the case, see Mercier *et al.* and Xu *et al.*). To the best of our knowledge, such strains are currently not available and would require extensive work to establish and characterise before the role of *mprF* gain-of-function mutations could be tested.

We agree that MprF's role in daptomycin resistance development was not sufficiently addressed. We have now expanded the discussion about daptomycin resistance, including the role of MprF (lines 275-282).

Mercier R, Kawai Y, Errington J. General principles for the formation and proliferation of a wall-free (L-form) state in bacteria. *Elife*. 2014 Oct 30;3:e04629.

Xu Y, Zhang B, Wang L, Jing T, Chen J, Xu X, Zhang W, Zhang Y, Han J. Unusual features and molecular pathways of *Staphylococcus aureus* L-form bacteria. *Microb Pathog*. 2020 Mar;140:103970.

Does growth phase affect the experimental outcomes? Cell wall inhibitors typically have much larger effects in log phase cultures, relative to stationary phase, whereas membrane disruption is typically less affected.

We thank the reviewer for this insightful comment. In response, we added new experimental data comparing daptomycin binding and induced membrane depolarisation between logarithmic and stationary growth phase cells. We observed that the membranes of stationary-phase cells bind daptomycin significantly less effectively than those of logarithmic-phase cells (new Fig. 5). However, when the daptomycin concentration is increased to compensate for the reduced binding, depolarisation is observed once again. Thus, daptomycin can indeed depolarise *B. subtilis* membranes even in a non-growing state. The reduced binding of daptomycin may be due to a decrease in the relative proportion of PG in *B. subtilis* membranes during the stationary growth phase (Giddeen *et al.*)
Giddeen J, Denson J, Liyanage R, Ivey DM, Lay JO. Lipid Compositions in Escherichia coli and Bacillus subtilis During Growth as Determined by MALDI-TOF and TOF/TOF Mass Spectrometry. Int J Mass Spectrom. 2009 Jun 1;283(1-3):178-184.

The daptomycin susceptibility of strains (MIC) should be provided to place findings in context. Previous work from this group showed dose-dependent effects of daptomycin on the inhibition of cell wall synthesis and membrane damage leading to lysis [reference 20 in the manuscript]. As such, the relationship between antibiotic concentration used and MIC needs to be made clear, especially since the authors seek to compare two different bacterial species.

We thank the reviewer for this recommendation. We have now included a table of daptomycin MICs for strains used in this study in the supplementary information (new Supplementary Table 1). We have also added the respective "multiple of MIC" -information to the text.

The *liaH*-deficient mutant does not form daptomycin-lipid clusters, which the authors propose contributes to the antibacterial activity of the antibiotic. This would suggest that the *liaH*-mutant is less susceptible to daptomycin. However, previous work has shown that a *liaH* mutant is more susceptible to daptomycin - how do the authors explain this?

In this manuscript, we do not provide evidence or suggest that daptomycin-induced lipid clusters contribute to the antibacterial activity; instead, we merely show that the IM30 homolog LiaH, and cell wall precursor lipids are essential for their formation. These domains have previously been hypothesised to contribute towards daptomycin's antibacterial activity (Pogliano *et al.*, Müller *et al.*), but our data with $\Delta liaH$ suggest that their formation is rather linked to cellular stress responses, and thus is consistent with slightly increased susceptibility. Importantly, these experiments do not imply that daptomycin is unable to interact with cell wall precursor lipids in the absence of LiaH, which is consistent with Grein *et al.*'s *in vitro* data and *in vivo* data generated in *S. aureus*, which does not encode for LiaH-homologs.

As part of an ongoing, larger collaboration, we are currently investigating how LiaH modulates membrane properties such as lipid domains, and how these translate to, or contribute to, protection from daptomycin. At this stage, it is too early to speculate.

Pogliano J, Pogliano N, Silverman JA. Daptomycin-mediated reorganization of membrane architecture causes mislocalization of essential cell division proteins. J Bacteriol. 2012 Sep;194(17):4494-504.

Müller A, Wenzel M, Strahl H, Grein F, Saaki TNV, Kohl B, Siersma T, Bandow JE, Sahl HG, Schneider T, Hamoen LW. Daptomycin inhibits cell envelope synthesis by interfering with fluid membrane microdomains. Proc Natl Acad Sci U S A. 2016 Nov 8;113(45):E7077-E7086. doi:

Grein F, Müller A, Scherer KM, Liu X, Ludwig KC, Klöckner A, Strach M, Sahl HG, Kubitscheck U, Schneider T. Ca²⁺-Daptomycin targets cell wall biosynthesis by forming a tripartite complex with

undecaprenyl-coupled intermediates and membrane lipids. Nat Commun. 2020 Mar 19;11(1):1455.

As an additional point for the role of LiaH, which is an interesting finding, the absence of complementation or comparison with other mutants reduces the strength of this finding.

We have now included new experimental data (Fig. S7) to compare LiaH to the other two membrane remodelling systems previously implicated in daptomycin susceptibility, MprF discussed above, and the lipid desaturase Des (<https://pubmed.ncbi.nlm.nih.gov/19164152/>). It turns out that, unlike LiaH, these systems are not required for the formation of daptomycin-induced fluid membrane areas, thus arguing for the specific role of LiaH in this process. This is now discussed in lines 232-234.

The authors provide evidence that the signal from sytox green is due to lysis triggered by daptomycin. However, a similar analysis is missing for *S. aureus*, which does not appear to lyse in the presence of daptomycin (supp figure 1). Instead, the authors employ assays to measure membrane depolarisation. As such, it is difficult to directly compare the effects of daptomycin on *B. subtilis* and *S. aureus*.

As recommended by the reviewer, we conducted a similar time-resolved SYTOX Green membrane permeability assay using a fluorescence plate reader in *S. aureus* (new Supplementary Fig. 2). These experiments show a delayed, highly heterogeneous increase in SYTOX Green fluorescence after daptomycin addition, which is consistent with the SYTOX Green fluorescence microscopy in Fig. 3b, and probably indicates heterogeneous, low-frequency autolysis that cannot be detected in OD measurements. Once again, the pore-forming lantibiotic nisin instead causes a rapid rise in SYTOX Green fluorescence in the plate reader, matching the DiSC₃(5) and SYTOX Green microscopy already presented in the manuscript.

Minor Points

Lines 30-31: please make clear that daptomycin has increased affinity relative to other bacterial phospholipids, otherwise 'increased affinity' lacks a comparator.

As suggested by the reviewer, we have now clarified this sentence by adding "compared to other bacterial phospholipid species".

New lines 31-32.

Line 59: please remove 'rather' as it reduces clarity as currently written.

We have removed this as recommended.

New line 60.

Line 76: Please replace 'at last' with 'finally'.

As suggested, we have replaced "at last" with "finally".

New line 77

Line 170. It should be mentioned that another study failed to replicate this finding (Kotsogianni et al., 2021). Whilst this may reflect different experimental approaches, it would help to make the authors' point that the role of lipid II was worth exploring further.

We thank the reviewer for bringing this study to our attention. We now mention these findings to highlight the importance of investigating daptomycin-lipid II interactions using independent approaches.

New Lines 31-32, 172-173.

Lines 187-188: I would be cautious here since the studies do not measure direct binding. It may be that the presence of lipid II has indirect effects via e.g. membrane fluidity.

The clustering of lipid II (or undecaprenyl-linked cell wall precursor in general) and its locally fluidising effect is indeed likely the reason why the observed domains are stained with fluidity-sensitive Nile Red. This is an area we have extensively studied in the context of MreB-delocalisation induced lipid domains, which also exhibit stronger Nile Red fluorescence (Strahl *et al.*). We are currently preparing another manuscript that details the link between Lipid II clustering and fluid lipid domains. We agree that our evidence in this manuscript falls short of demonstrating a direct interaction, which is why we argue that our experiments *support* the interaction model. We have rephrased the sentence to clarify this point (Importantly, this phenomenon provides indirect *in vivo* support for the interaction between daptomycin and cell wall precursor lipids).

New lines 190-191.

Strahl H, Bürmann F, Hamoen LW. The actin homologue MreB organizes the bacterial cell membrane. *Nat Commun.* 2014 Mar 7;5:3442.

Line 193: why use gramicidin and not nisin here, as used earlier in the manuscript?

The pore formation by nisin relies on interaction and polymerisation with lipid II (Breukink & de Kruijff). Consequently, it cannot be used against L-forms lacking cell wall precursor lipids. Instead, gramicidin was used, which is a monovalent cation channel-forming peptide that is not dependent on the presence of cell wall precursors (Hladky & Haydon).

Breukink E, de Kruijff B. Lipid II as a target for antibiotics. *Nat Rev Drug Discov.* 2006 Apr;5(4):321-32.
Hladky SB, Haydon DA. Ion transfer across lipid membranes in the presence of gramicidin A. I. Studies of the unit conductance channel. *Biochim Biophys Acta.* 1972 Aug 9;274(2):294-312.

Figure 1e

Figure 1e

Figure 2c

Figure 4c

L-forms (- lipid II)

value	count
0	8213
1	0
2	0
3	0
4	0
5	0
6	0
7	0
8	0
9	0
10	0
11	0
12	0
13	0
14	0
15	0
16	0
17	1535
18	0
19	0
20	0
21	0
22	0
23	0
24	0
25	0
26	252
27	0
28	0
29	0
30	0
31	0